# Field Theoretical Approach for Signal Detection in Nearly Continuous Positive Spectra II: Tensorial Data

**DOI:** 10.3390/e23070795

**Published:** 2021-06-23

**Authors:** Vincent Lahoche, Mohamed Ouerfelli, Dine Ousmane Samary, Mohamed Tamaazousti

**Affiliations:** 1Université Paris-Saclay, CEA, List, F-91120 Palaiseau, France; vincent.lahoche@cea.fr (V.L.); mohamed-oumar.ouerfelli@cea.fr (M.O.); mohamed.tamaazousti@cea.fr (M.T.); 2International Chair in Mathematical Physics and Applications (ICMPA-UNESCO Chair), University of Abomey-Calavi, Cotonou 072B.P.50, Republic of Benin

**Keywords:** renormalization group, field theory, phase transition, random tensors, big data, principal component analysis, tensorial principal component analysis, signal detection, 05.10.Cc, 05.40.-a, 29.85.Fj

## Abstract

The tensorial principal component analysis is a generalization of ordinary principal component analysis focusing on data which are suitably described by tensors rather than matrices. This paper aims at giving the nonperturbative renormalization group formalism, based on a slight generalization of the covariance matrix, to investigate signal detection for the difficult issue of nearly continuous spectra. Renormalization group allows constructing an effective description keeping only relevant features in the low “energy” (i.e., large eigenvalues) limit and thus providing universal descriptions allowing to associate the presence of the signal with objectives and computable quantities. Among them, in this paper, we focus on the vacuum expectation value. We exhibit experimental evidence in favor of a connection between symmetry breaking and the existence of an intrinsic detection threshold, in agreement with our conclusions for matrices, providing a new step in the direction of a universal statement.

## 1. Introduction

In several areas of physics where numerous interacting degrees of freedom are expected, we are aiming to extract relevant features at large scales for microscopic physics. From a general point of view, the collective behavior of a large number of elementary degrees of freedoms, like molecules in a fluid, can be suitably described by an effective partial differential equation, replacing the very large number of data required to describe exactly the state of the fluid [1,2]. At a very large scale with respect to the molecular size, the knowledge of the exact positions and velocities of each molecule is equivalent to the knowledge of a velocity field $\vec{v}\left(x,y,z,t\right)$ with four components depending on space and time, which are the solution of the Navier–Stokes equation [3,4]. Another standard example is provided by the behavior of the Ising model in the vicinity of the ferromagnetic transition, where the behavior of the discrete spins taking values S=±1 are well described by an effective continuous scalar field ϕ(x,y,z). The renormalization (semi-)group (RG) concept is probably the most important discovery of the XXth century to explain the emergence and the universality of large scale effective physics. Note that the modern incarnation of the RG due to Kadanoff and Wilson [5,6,7,8,9,10,11] takes the form of a flow into the space of allowed Gibbs measures, obtained from a partial integration procedure of the degrees of freedom below some running scale, and that provides an effective description for the remaining degrees of freedom. The running scale discriminates between the microscopic states, that we ignore, and the macroscopic states, that we keep. The advantage of this point of view is that a complete description of the fundamental constituents of matter is not required, since microscopic physics are absorbed in the parameter defining the effective description for large scale degrees of freedom. With this respect, what is relevant in this description is the behavior of these parameters along with the flow; some of them tend to turn-off and others intensify.

Data sciences provide a non-conventional area of application for RG. Indeed, data analysis aims to deal with very large sets of data that have non-trivial correlations. Principal component analysis (PCA) [12,13,14,15,16,17,18,19] is one of the standard tools in data analysis allowing to extract relevant features from such large data sets. It consists of performing a suitable linear projection onto a lower-dimensional space, corresponding to the eigen directions associated with the largest eigenvalues of the covariance matrix. In favorable cases, a few numbers of eigenvalues provide a sharp separation between important and unimportant features. For nearly continuous spectra, however, standard PCA fails to provide a clean separation, and the RG group has been considered as a promising tool to investigate these challenging issues. The analogy between PCA and RG has been taken to a more formal level in a few recent papers. A connection through information theory has been discussed in [18,20,21], where the authors focus on the ability to distinguish effective states from the progressive information dilution due to coarse-graining. A field theoretical embedding has been firstly introduced in [17], and continued in [22,23,24] in a nonperturbative framework with a complete analogy of what happens in standard field theory. The central object that connects these approaches is the Fisher information metric, which is none other than an infinitesimal version of the Kullback–Leibler divergence (or mutual entropy) [25,26] (Note that our field theories are law exponentials, and the Fisher metric is identified with the second cumulant, on which the main part of our investigation is based). Indeed, it is well known that the dimension of the relevant sector in the theory space, spanned by relevant and marginal couplings, depends on the dimension *d* of the space, or more specifically, on the momenta distribution $\rho \left(p^{2}\right)={\left(p^{2}\right)}^{d/2-1}$. In the same direction, the following question becomes relevant: *what kind of RG can be supported by the distribution of the eigenvalues of the covariance matrix, in a field theoretical framework able to reproduce (at least partially) the data correlations, and extract relevant features of these distributions*? Note that such a strategy is precisely the way field theories are understood nowadays—as effective descriptions of incomplete knowledge of the true microscopic reality [27]. More generally, other approaches based on RG have been successfully used to address various important problems in artificial intelligence [28,29].

Let us provide the following important conclusions of our previous investigations before starting this paragraph. In references [22,23,24], we come to the following two significant findings. The first one is about the relevance of interactions; more specifically, the dimension of the relevant sector of the (minimal) theory space is reduced by the presence of a signal in comparison to its expected value for Marchenko–Pastur law [30]. The second one is that the presence of a signal in the spectrum can be materialized as $Z_{2}$-symmetry breaking. These results focus on data sets which can be suitably described by a (suitably mean-shifted) large N×P matrix $X_{ia}$, for i=1,⋯,N and a=1,⋯,P, and a covariance matrix C corresponding to the average of the product $XX^{T}$. In this paper, we propose to go beyond these results and consider data sets suitably described by tensors (multidimensional arrays) $X_{ia_{1}a_{2}\cdots a_{k-1}}$, i.e., cubes or hyper-cubes rather than tables of numbers. Tensorial principal component analysis (TPCA) (also referenced as multilinear principal component analysis (MPCA) in the literature) [31,32,33,34,35,36,37] is an extension of the standard PCA that aims to recover or estimate a signal merged into a noise. The difficulty comes from the fact that many powerful tools used in standard PCA do not work for tensors. In particular, the notions of eigenvalues and eigenvectors, which are essential for the matrix PCA, not only lack a clear and unique generalized definition for the tensors [38,39], but also computing them becomes NP-hard [40]. The difference of difficulty between matrix PCA and tensor PCA can be observed even in the simplest case of tensor PCA. This case consists of one spike $u^{⊗k}\in {\left(R^{N}\right)}^{⊗k}$ associated to a normalized vector $u\in R^{N}$, corrupted by a noise tensor $Z\in {\left(R^{N}\right)}^{⊗k}$ whose components $Z_{i_{1}\cdots i_{k}}$ are i.i.d Gaussian random variables. Thus, the tensor from which we aim to recover the signal is given by $X=Z+\lambda u^{⊗k}$, where λ is the signal to noise ratio. In this model, it has been proven using information theory tools that it is theoretically possible to recover or detect the spike above the theoretical threshold $\lambda_{\mathrm{th}}=O\left(N^{1/2}\right)$, but, interestingly, there is no known tractable algorithm (with a polynomial-time complexity) that can do so below an algorithmic threshold estimated at $\lambda_{\mathrm{alg}}=O\left(N^{k/4}\right)$. For $\lambda_{\mathrm{th}}\leq \lambda \leq \lambda_{\mathrm{alg}}$, solving tensor PCA is expected to be hard. The hardness of the problem can be inferred from the existence of an energy landscape with exponential complexity, containing many uninformative critical points. Recovering the signal vector amounts to finding the global maximum of the function $f:\;u\mapsto \langle X,\;u^{⊗k}\rangle$ (where ${⊗}^{k}$ denotes the *k*th tensorial product: $u^{⊗k}=\underset{k\;\mathrm{times}}{\underset{︸}{u⊗\cdots ⊗u}}$). Thus, the hardness of the problem can be attributed to the high dimensional and highly non-convex landscape associated with *f*. Indeed, using tools inspired by statistical physics, it has been shown that the number of critical points orthogonal to the direction of the signal is exponentially large in *N* [33]. These uninformative minima are suspected to be the reason for the failure of the algorithms below the algorithmic threshold.

In this paper, as a continuation of [22,23,24], we propose to use an effective field theoretical embedding able to partially reproduce the correlations to address some issues in TPCA through a suitable coarse-graining procedure to construct RG flow. The modes over which partial integration is defined are the eigenvalues of a slight generalization of the standard covariance matrix, whose entries $C_{ij}$ are the averaging of $\sum_{a_{1},\cdots ,a_{k-1}}X_{ia_{1}\cdots a_{k-1}}X_{ja_{1}\cdots a_{k-1}}$. This quantity has been recently considered to construct promising TPCA algorithms [37] based on tensorial invariants coming from random tensor models [41,42,43,44]. For this reason, we expect its spectrum to be a good candidate for coarse-graining approaches. As in [22,23,24], we focus on nearly continuous spectra, where standard PCA tools fail to provide a clean separation between the relevant features that we aim to keep and irrelevant ones that we want to ignore. We consider an effective probability distributions for a *N*-component field $\varphi_{i}\in R$. Although such a field theoretical embedding is certainly simplified, we think the lessons that can be learned from its investigations provide instrumental guidelines toward a true description. Indeed, we do not claim that the effective behavior of this field can be directly related to specific details on the data. However, we may expect that the global properties of field distribution, reflecting those of the fundamental modes, provide some objective criteria to decide if data are purely noisy or not. In particular, we show in this paper that the conclusions of [22,23,24] for the matrix model hold for the tensor case, and therefore:(i)The dimension of the relevant sectors of the theory space decreases for a strong enough signal, consequently providing a first objective criterion to define the signal detection threshold.(ii)The presence of a signal in the spectra may be revealed by a $Z_{2}$-symmetry breaking for the effective distribution. This provides a second intrinsic detection threshold. A strong enough signal is required to change the shape of the effective potential and therefore the end vacuum expectation value for the classical field $\langle \varphi_{i}\rangle$.

## 2. Preliminaries

In this section, we introduce the theoretical material required for the numerical investigations of Section 3. We introduce the tensorial formalism and discuss the different generalizations of the covariance matrix in this context. Then, following [22,23,24], we aim to construct an effective field theory able to reproduce at least partially the data correlations and investigate some approximation of the exact equation describing its intrinsic RG flow, focusing on standard local potential approximation (LPA). As a result that the equations and arguments are essentially the same as in the previously cited papers, we sketch the discussion as the reader may find an extended version in these references.

### 2.1. Some Basics of Framework

As explained in the introduction, we focus on data sets which can be essentially described by *k*-dimensional hyper-cubes of numbers, suitably represented as a real tensor $X:E_{1}⊗E_{2}\cdots ⊗E_{k}\mapsto R$ of rank *k* with entries $X_{i_{1}i_{2}\cdots i_{k}}\in R$, and $E_{\alpha }\subset N$ with cardinality $\#E_{\alpha }=:P_{\alpha }$. With matrices, for k=2, the 2-point correlations are quantified with the entries of the covariance matrix C. It is defined as the averaging of $XX^{T}$, and the size of its eigenvalues provides a quantitative measure of relevance. With respect to this, a spectral analysis that retains only the most relevant features, for example by providing the best rank-one approximation for *X*, is one of the most popular techniques to “denoise” a signal. A first difficulty for the generalization from the matrix to tensors is that there is no a priori single candidate to play the role of the covariance matrix. Indeed, for a matrix *X*, the N×N quantities $\sum_{a}X_{ia}X_{ja}\equiv X_{i}\cdot X_{j}$ are the only connected (Obviously any product of O(P) invariant is invariant as well. A connected invariant is an invariant which is not itself a product of invariants) O(P)-invariants that we can build from *N* vectors of length *P* (we recall that O(n) designates orthogonal transformations acting on vectors of length *n*: $o\in O\left(n\right)\to oo^{T}=o^{T}o=\mathrm{id}$; including rotations and reflections), $X_{i}=\left\{X_{ia}\right\}$. In contrast, the situation is not so suitable for tensors. Indeed, if we consider only matrix-like correlations with entries $C_{ij}$, the simplest connected $\prod_{\alpha =2}^{k}O\left(P_{\alpha }\right)$-invariant object that we can think to generalize $XX^{T}$ is:(1)$$C_{ij}=\langle \sum_{a_{2}\cdots a_{k}=1}^{N}X_{ia_{2}\cdots a_{k}}X_{ja_{2}\cdots a_{k}}\rangle ,$$
suitably averaged as the notation 〈Q〉 indicates. Following the terminology of random tensor model [42], we call *elementary melonic approximation* this object. We recall that melons in the 1/N expansion of the random tensor models correspond to the leading order contributions, the elementary one being constructed with two copies of the random tensor. Indeed, we can also consider a more complicated object, crossing indices between a larger number of copies of the tensor *X*. For instance, assuming that each index is independent of the (k−1) others, we may have four copies of the tensor *X*.
(2)$$\begin{array}{cc} C_{ij}^{\prime }= & \langle \sum_{\left\{a_{l}\right\},\left\{a_{l}^{\prime }\right\}}^{N}X_{ia_{2}\cdots a_{l}a_{l+1}\cdots a_{k}}X_{ja_{2}\cdots a_{l}a_{l+1}^{\prime }\cdots a_{k}^{\prime }}X_{a_{1}^{\prime }a_{2}^{\prime }\cdots a_{l}^{\prime }a_{l+1}^{\prime }\cdots a_{k}^{\prime }}X_{a_{1}^{\prime }a_{2}^{\prime }\cdots a_{l}^{\prime }a_{l+1}\cdots a_{k}}\rangle . \end{array}$$
All of these objects can be constructed from $\prod_{\alpha =1}^{k}O\left(P_{\alpha }\right)$-invariants considered in random tensor models [41,42,44]. These tensorial invariants can be nicely labeled with *k*-colored regular graphs (see Figure 1) as follows: we associate a vertex to each tensor X, the *k* colored edges hooked to it correspond to the *k* indices of the tensor. Then, each of the edges is connected following their respective colors, i.e., following the contraction of the indices in the tensor invariant. Different definitions for the matrix C can thus be obtained by opening one of these colored edges. Such a generalization of the covariance matrix has been considered recently [37] as a novel and promising tool in the investigation for tensorial PCA. Indeed, it was proven that similarly to the matrix case, the leading eigenvector of these covariance matrices becomes highly correlated with the signal vector (corresponding to the relevant features) above a given threshold. A sharp separation between the eigenvalues that are related to relevant features and the irrelevant ones has also been observed when the number of signal vectors is small. This renders the use of these covariance matrices natural for the investigation of nearly continuous spectra in the tensor case.

Throughout this manuscript, we focus on the simplest definition given in Equation (Equation 1), whose relevance has been pointed out by the authors of [37], and whose spectrum is positive definite. Moreover, we focus on *hyper-cubic tensors*, imposing $P_{\alpha }=N\;\forall \alpha$. Figure 2 illustrates a typical spectrum for large *N*, for purely noisy data (on the top) materialized as a random tensor with i.i.d Gaussian entries, and with a signal built as a sum of spikes (on the bottom). The first histogram is obtained from 100 realizations of the eigenvalue distribution associated to i.i.d random tensors of rank 3 and size $N^{3}=50^{3}$. The second picture is constructed from a sum of a random tensor and 50 (suitably normalized) spikes. Finally, the green curve materializes the numerical interpolation.

The simplest incarnation of field theoretical embedding that we can think of focuses on a set of *N* random variables $\Phi :=\left\{\phi_{i}\right\}\in R^{N}$ with a purely Gaussian distribution:(3)$$p\left[\Phi \right]\propto \exp\left(\frac{1}{2}\;\sum_{i,j=1}^{N}\phi_{i}C_{ij}^{-1}\phi_{j}\right),$$
for which the 2-point correlation is precisely $\bar{\phi_{i}\phi_{j}}=C_{ij}$ from construction. The notation $\bar{x}$ indicates an averaging with respect to the probability distribution p[Φ] of the quantity *x*. Note that for this expansion, we assume that the random variables $\phi_{i}$ are distributed with zero means for simplicity. The standard Wick theorem ensures that all the momenta of the distribution decompose as a product of the variance $C_{ij}$, so that cumulants (the one particle irreducible (1PI) *n*-points correlation function in the field theory language) higher than two vanish. Thus, more general distributions, describing correlations with more than 2-points, require to retain non-Gaussian terms in the classical Hamiltonian (the log-likelihood in probability theory). As an example:(4)$$H\left[\Phi \right]=\frac{1}{2}\;\sum_{i,j=1}^{N}\phi_{i}{\tilde{C}}_{ij}^{-1}\phi_{j}+g\sum_{i}\phi_{i}^{4}+\cdots ,$$
where we introduced the “*tilde notation*” for ${\tilde{C}}_{ij}$ (the free propagator) to avoid confusion with $C_{ij}$. The quartic truncation around $\phi^{4}$ interactions “at the same point” *i* has been considered in [17,22] as a minimal model beyond purely Gaussian distribution. Note that because we expect to recover the empirical correlations $C_{ij}$ from p[Φ], ${\tilde{C}}_{ij}^{-1}$ cannot be identified with $C_{ij}^{-1}$. Indeed, the two matrices should be equal at the first order in perturbation theory around the Gaussian point, but differ when we take into account quantum corrections:(5)$$C_{ij}={\tilde{C}}_{ij}+O\left(g\right).$$
Inferring the explicit expression for ${\left(\tilde{C}\right)}^{-1}$ from the knowledge of $C^{-1}$ is a difficult task even for the simpler field theories [45,46]. RG is non-invertible by construction: the microscopic pieces of information are lost from coarse-graining and many microscopic models can have the same large scale behavior. A strong simplification is to estimate the difference between the two matrices as a translation of *all* the eigenvalues by a constant *k*, taking into account the most relevant quantum effects:(6)$$C_{ij}^{-1}\approx {\tilde{C}}_{ij}^{-1}+k.$$
The LPA that we will consider in this paper to construct solutions of the exact RG equation is compatible with this assumption. Indeed, LPA focuses essentially on the infrared aspects, and affects mainly the lowest eigenvalue of the ${\tilde{C}}_{ij}^{-1}$ spectrum. In such a way, the respective eigenvalue distributions of the two matrices ${\left(\tilde{C}\right)}^{-1}$ and $C^{-1}$ are expected to be the same, up to a global translation by the difference between their smallest eigenvalues. We denote as $p^{2}$ these rescaled eigenvalues, positive from construction, and as $\rho (p^{2})$ their distribution. Formally:(7)$$\rho \left(x\right)\equiv \frac{1}{N}\sum_{\mu =1}^{N}\delta \left(x-p_{\mu }^{2}\right),$$
where δ(x) is the ordinary Dirac distribution and the index μ labels the eigenvalues. The choice of the monomials spanning the second part of the classical Hamiltonian and containing non-Gaussian terms is not guided a priori. As discussed in [17,22], we focus on the simplest case of local interactions in the usual sense in field theory, based on monomials of the form $\phi_{i}^{n}$. We focus on the $Z_{2}$-symmetric case, an additional but inessential simplification allowing to treat positive and negative fluctuations on an equal footing. Thus, n=2p for p∈N.

For convenience, regarding RG investigations, it is suitable to work in the eigenbasis of the matrix ${\left(\tilde{C}\right)}^{-1}$, in which our assumption is the same as for $C^{-1}$. In that way, the Gaussian (or kinetic) part of the classical Hamiltonian takes the form:(8)$$H_{\mathrm{kinetic}}\left[\psi \right]=\frac{1}{2}\sum_{\mu =1}^{N}\psi_{\mu }\lambda_{\mu }\psi_{\mu },$$
where $\lambda_{\mu }$ denote the eigenvalues of ${\tilde{C}}^{-1}$, labeled with the discrete index μ; and the fields $\left\{\psi_{\mu }\right\}$ are the eigen-components of the expansion of $\phi_{i}$ along the normalized eigenbasis $u_{i}^{\left(\mu \right)}$:(9)$$\psi_{\mu }=\sum_{i=1}^{N}\phi_{i}u_{i}^{\left(\mu \right)},\;\sum_{j}{\tilde{C}}_{ij}^{-1}u_{j}^{\left(\mu \right)}=\lambda_{\mu }u_{i}^{\left(\mu \right)}.$$
Denoting as $m^{2}$ the smallest eigenvalue, we have, from the definition of $p_{\mu }^{2}$, $\lambda_{\mu }=p_{\mu }^{2}+m^{2}$. As a result, the kinetic Hamiltonian takes formally the form of the standard kinetic Hamiltonians in field theory:(10)$$H_{\mathrm{kinetic}}\left[\psi \right]=\frac{1}{2}\sum_{\mu =1}^{N}\psi_{\mu }\left(p_{\mu }^{2}+m^{2}\right)\psi_{\mu }.$$
It is, however, more difficult to deal with interactions using this formalization. In [22,23,24], we simplified the problem by doubling the number of eigenvalues, introducing a momentum-dependent field ψ(p), having the same variance as the original field $\psi_{\mu }$. The eigenvalues *p* are positive or negative, but their square $p^{2}$ is distributed following the empirical distribution $\rho (p^{2})$. In such a way, it is suitable to define locality of the interaction directly in the momentum space, working with conservative interactions in the usual sense in field theory; and the interaction part of the classical Hamiltonian (which retain monomials of degree higher than 2) can be written as:(11)$$H_{\mathrm{int}}\left[\psi \right]=\sum_{P=1}^{\infty }u_{2P}\sum_{\left\{p_{\alpha }\right\}}\;\delta_{0,\sum_{\alpha =1}^{2P}p_{\alpha }}\;\prod_{\alpha =1}^{2P}\;\psi \left(p_{\alpha }\right),$$
where δ denotes the standard Kronecker delta. These simplifications are expected to be inessential for our considerations, focusing on global aspects of distributions at a large scale. According to the usual definition in physics, we call large scale, or infrared scale (IR), the sector of the small eigenvalues of the propagator (large eigenvalues of the covariance matrix). The opposite limit is similarly called ultraviolet (UV).

From this field theory, we aim to construct nonperturbative renormalization group *a la Wilson*, based on the integrating-out of momentum modes from a path-integral representation of the theory [5,6,7,8,9,10,11,47]. We make use of the functional renormalization group (FRG) formalism, based on the Legendre transform Γ[M] of the free energy (the functional generating of cumulants of the distribution p[ψ]) $W\left[j\right]:=\ln\left(\int d\psi \;p\left[\psi \right]e^{\sum_{p}j\left(-p\right)\psi \left(p\right)}\right)$, which can be viewed as an effective Hamiltonian at large scale, taking all fluctuations’ effects. FRG is reputed for its flexibility in regards to models having strong correlations and large coupling, and for avoiding the well-known instabilities occurring for truncations working with the effective Hamiltonian for the remaining degrees of freedom [48]. Moreover, optimization techniques are available to control the physical RG flow within systematic approximations [49,50]. FRG focuses on the *effective averaged Hamiltonian* $\Gamma_{k}$, which can be interpreted as the effective Hamiltonian of the integrated-out degrees of freedom, and interpolates between the microscopic classical Hamiltonian H for k≫1 and the full effective hamiltonian Γ for k=0. The Wetterich–Morris equation describes how $\Gamma_{k}$ is modified from infinitesimal integrating-out of momenta $p^{2}$ in a small neighborhood around $p^{2}=k^{2}$. It writes in this context [51,52]:(12)$${\dot{\Gamma }}_{k}=\frac{1}{2}\;\int dpp\rho \left(p^{2}\right){\dot{r}}_{k}\left(p^{2}\right){\left(\Gamma_{k}^{\left(2\right)}+r_{k}\right)}_{p,-p}^{-1},$$
where $\dot{x}:=kdx/dk$. The function $r_{k}\left(p^{2}\right)$ is the regulator in the momentum representation. It is such that $r_{k}\left(p^{2}\right)\to 0$ for $k^{2}/p^{2}\to 0$, $r_{k}\left(p^{2}\right)\to k^{2}$ for k→Λ, and $r_{k}\left(p^{2}\right)>0$ for $p^{2}/k^{2}\to 0$. Here, Λ≫1 denotes some fundamental ultraviolet cut-off. The stability and the convergence of the RG flow depend on the choice of the regulator [49,50,53]. Furthermore, $\Gamma_{k}^{\left(2\right)}$ denotes the second functional derivative of $\Gamma_{k}\left[M\right]$ with respect to the classical field *M*. We recall that $\Gamma_{k}$ is defined as the Legendre transform of the quantity $W_{k}\left[j\right]-\frac{1}{2}\sum_{p}r_{k}\left(p^{2}\right)M\left(p\right)M\left(-p\right)$; the free energy $W_{k}\left[j\right]$, at scale *k*, being defined as:$$e^{W_{k}\left[j\right]}:=\int d\psi p\left[\psi \right]e^{\sum_{p}j\left(-p\right)\psi \left(p\right)-\frac{1}{2}\sum_{p}r_{k}\left(p^{2}\right)\psi \left(p\right)\psi \left(-p\right)}.$$
The exact RG Equation (Equation 12) works in an infinite-dimensional functional space, and solving this equation is a difficult task even for the simplest models. Only approximate solutions can be obtained, in practice, by truncating the flow into a finite subregion of the full theory space of the allowed Hamiltonians. In this paper, we focus on the popular LPA approximation, which we will now briefly introduce.

### 2.2. Local Potential Approximation

LPA is among the most popular truncations considered in the literature [52,54,55,56,57,58]. The heart of this approximation is to keep only the first term in the momenta expansion (called “derivative expansion” in the literature) of the full effective Hamiltonian $\Gamma_{k}$. In particular, the momentum dependence of the classical field M(p) is assumed to be dominated by the zero-momentum (large scale) value:(13)$$M\left(p\right)∼m\delta_{p0}=:M_{0}.$$
We thus define the effective potential $U_{k}$ as $\Gamma_{k}\left[M=M_{0}\right]=:NU_{k}\left[\chi \right]$; which, in turn, is approximated as:(14)$$U_{k}\left[\chi \right]=\frac{u_{4}\left(k\right)}{2!}{\left(\chi -\kappa (k)\right)}^{2}+\frac{u_{6}\left(k\right)}{3!}{\left(\chi -\kappa (k)\right)}^{3}+\cdots ,$$
with $\chi :=m^{2}/2$, and κ(k) denotes the running vacuum. The 2-point vertex $\Gamma_{k}^{\left(2\right)}$ is defined, again at first order in the momenta expansion, as:(15)$$\Gamma_{k,\mu \mu^{\prime }}^{\left(2\right)}=\left(Z\left(k\right)p^{2}+\frac{\partial^{2}U_{k}}{\partial M^{2}}\right)\delta_{p_{\mu },-p_{\mu^{\prime }}},$$
The flow equation for $U_{k}$ can be deduced from (Equation 12), setting $M=M_{0}$ on both sides. We get:(16)$${\dot{U}}_{k}\left[M\right]=\frac{1}{2}\;\int pdp\;k\partial_{k}\left(r_{k}\left(p^{2}\right)\right)\rho \left(p^{2}\right)\left(\frac{1}{\Gamma_{k}^{\left(2\right)}+r_{k}}\right)\left(p,-p\right).$$
In definition (Equation 15), we introduce the anomalous dimension Z(k), which has a non-vanishing flow equation for κ≠0. To take into account the non-vanishing flow for *Z*, it is suitable to factorize a global *Z* factor in front of the definition of $r_{k}$. We choose to work with the optimized Litim regulator, which has been proved to have nice properties in regards to optimization, stability and integrability [59]:(17)$$r_{k}\left(p^{2}\right)=Z\left(k\right)\left(k^{2}-p^{2}\right)\theta \left(k^{2}-p^{2}\right).$$
Note that this factorization is not inoffensive, and may introduce disagreements with the required boundary conditions [53,60,61,62,63]. We distinguish two levels of approximation, the standard LPA where we enforce Z(k)=1, and the improved version LPA${}^{\prime }$, taking into account the RG flow of the running field strength Z(k). However, in [23], the authors showed that taking into account the field strength renormalization provides only a slight change in the behavior of the RG flow. Our numerical investigations for tensors confirm that it plays again a minor role, and we focus on standard LPA for this paper.

As a first step and following [22,23], we introduce the new flow parameter:(18)$$\tau :=\ln\left(2\int_{0}^{k}\rho \left(p^{2}\right)pdp\right),$$
and after some algebra we arrive at the expression:(19)$$U_{k}^{\prime }\left[\chi \right]=k^{2}\rho \left(k^{2}\right){\left(\frac{dt}{d\tau }\right)}^{2}\;\frac{k^{2}}{k^{2}+\partial_{\chi }U_{k}\left(\chi \right)+2\chi \partial_{\chi }^{2}U_{k}\left(\chi \right)},$$
with the notation $x^{\prime }:=dx/d\tau$. In standard applications of the RG flow, the running quantities are suitably rescaled to transform RG equations in an autonomous system. For general distributions, such rescaling does not hold (more details may be found in [52]). However, it is suitable to rescale the parameters in such a way that only the linear terms are not autonomous. These terms correspond generally to the ones for which we have a contribution of the canonical dimensions. In such a way, following [23], we define the scaling of the effective potential as:(20)$$\partial_{\chi }U_{k}\left(\chi \right)k^{-2}=\partial_{\bar{\chi }}{\bar{U}}_{k}\left(\bar{\chi }\right),\;\chi \partial_{\chi }^{2}U_{k}\left(\chi \right)k^{-2}=\bar{\chi }\partial_{\bar{\chi }}^{2}{\bar{U}}_{k}\left(\bar{\chi }\right),$$
and we obtain:(21)$$U_{k}^{\prime }\left[\chi \right]={\left(\frac{dt}{d\tau }\right)}^{2}\;\frac{k^{2}\rho \left(k^{2}\right)}{1+\partial_{\bar{\chi }}{\bar{U}}_{k}\left(\bar{\chi }\right)+2\bar{\chi }\partial_{\bar{\chi }}^{2}{\bar{U}}_{k}\left(\bar{\chi }\right)}$$
The definitions in (Equation 20) fix the *relative scaling* between $U_{k}$ and χ. Moreover, the previous relation requires for $U_{k}$ the following rescaling:(22)$$U_{k}\left[\chi \right]:={\bar{U}}_{k}\left[\bar{\chi }\right]k^{2}\rho \left(k^{2}\right){\left(\frac{dt}{d\tau }\right)}^{2},$$
and the corresponding rescaling for χ may be straightforwardly deduced from (Equation 20):(23)$$\chi =\rho \left(k^{2}\right){\left(\frac{dt}{d\tau }\right)}^{2}\bar{\chi }.$$
This equation defines the dimension of κ, which has to be the same as the one of χ. Furthermore, the flow equations for the different couplings can be easily derived from the parameterization of $U_{k}$: (24)$$\frac{\partial U_{k}}{\partial \chi^{n}}{|}_{\chi =\kappa }=u_{2n}\left(1-\delta_{1n}\right).$$
Finally, working at fixed $\bar{\chi }$ rather than fixed χ, we deduce the final expression for the dimensionless flow equation:(25)$$\begin{array}{cc} {\bar{U}}_{k}^{\prime }\left[\bar{\chi }\right]= & -\dim_{\tau }\left(U_{k}\right){\bar{U}}_{k}\left[\bar{\chi }\right]+\dim_{\tau }\left(\chi \right)\bar{\chi }\frac{\partial }{\partial \bar{\chi }}{\bar{U}}_{k}\left[\bar{\chi }\right]+\frac{1}{1+\partial_{\bar{\chi }}{\bar{U}}_{k}\left(\bar{\chi }\right)+2\bar{\chi }\partial_{\bar{\chi }}^{2}{\bar{U}}_{k}\left(\bar{\chi }\right)}. \end{array}$$
The explicit expressions for the *canonical dimensions* $\dim_{\tau }\left(U_{k}\right)$ and $\dim_{\tau }\left(\chi \right)$ are:(26)$$\dim_{\tau }\left(U_{k}\right)=t^{\prime }\frac{d}{dt}\ln\left(k^{2}\rho \left(k^{2}\right){\left(\frac{dt}{d\tau }\right)}^{2}\right),$$
and
(27)$$\dim_{\tau }\left(\chi \right)=t^{\prime }\frac{d}{dt}\ln\left(\rho \left(k^{2}\right){\left(\frac{dt}{d\tau }\right)}^{2}\right).$$
These canonical dimensions set in turn the dimensions of all the couplings. Denoting as ${\bar{u}}_{2n}$ these dimensionless couplings, and $\beta_{2n}:={\bar{u}}_{2n}^{\prime }$, the canonical dimension of $u_{2n}$ is defined as the term of order zero of $\beta_{2n}/{\bar{u}}_{2n}$.

## 3. Flow Equations and Numerical Investigations

In this section, we consider numerical investigations of the RG flow equations in the LPA. Note that in contrast to the matrix model case, we do not have an analytic formula, analogous to the Marchenko–Pastur law, for the eigenvalue distribution μ(λ) of the covariance matrix. For this reason, all of our analysis will be completely numerical and based on the interpolations of histograms such as the ones provided in Figure 2.

First, we consider the behavior of the canonical dimensions for local couplings $u_{2n}$, defined from Equation (Equation 24). In Figure 3, the top left plot corresponds to the opposite of the canonical dimensions for the first odd local interactions for $\varphi^{4}$ (cyan curve), $\varphi^{6}$ (green curve), $\varphi^{8}$ (blue curve), and $\varphi^{10}$ (magenta curve) associated to the eigenvalue distribution corresponding to one of the possible generalizations of the covariance matrix for a purely random tensor (red curve). The bottom left plot corresponds to the canonical dimensions associated to the eigenvalue distribution of the generalized covariance matrix of the same random tensor but with a signal (some spikes added to it). The right plots are a zoom of the respective left plots. Note that this figure reports an illustration of typical results that we obtain, from purely noisy spectra, and when a signal is added, built as a superposition of a large number of spikes. Let us focus first on the diagrams for pure noise (the two diagrams on the top): up to a certain eigenvalue on the spectrum, the value of the dimension for $u_{6}$ tends to vanish. This situation is very reminiscent of what happens for matrices, but several differences should be noted. First of all, the marginal behavior of $u_{6}$ appears much earlier than for matrices and tends to be maintained. This is also the case for all canonical dimensions. All of them end up reaching a plateau from which they hardly move any more, unlike what has been observed for the matrices, where the dimensions continued to change at all scales. Thus, the behavior of the flow, in this case, corresponds almost exactly to that of an ordinary field theory, where the canonical dimensions are fixed once and for all by the dimension of the reference space, and the flow equations tend to be reduced to an autonomous system, admitting fixed point solutions. The last point of difference with the flow associated with the law of Marchenko–Pastur comes from the behavior of the canonical dimensions in the UV, on the side of the small eigenvalues. While in the matrix case these dimensions tend to become very positive, and this trend concerned an arbitrarily large number of couplings, we observe exactly the opposite behavior in the case of tensors. The number of relevant interactions gradually decreases, and all non-Gaussian couplings tend to become irrelevant within the UV limit. For all these reasons, we expect the field theory approximation to perform better in the tensor case rather than in the matrix model.

In Figure 4, we provide a numerical integration of the flow equations corresponding to a quartic truncation in the parameterization:(28)$$U_{k}\left[\chi \right]=u_{2}\chi +\frac{1}{6}u_{4}\chi^{2}+\frac{1}{90}u_{6}\chi^{3}+\cdots ,$$
assuming a suitable expansion around χ=0. The domain of validity for this approximation is called *symmetric phase*, and despite that we do not limit our investigations to this region and that our formalism allows us to go beyond this approximation, it is instructive to start in this regime. The resulting pictures are very reminiscent of what happens in critical phenomena. In both pictures we observe the existence of a quasi-fixed point which separates the flow into two different regions—one with positive masses and another with negative masses. Although it is difficult to interpret separately one or the other of these diagrams, their superposition is quite instructive. Indeed, the position of the quasi-fixed point changes when a signal is added to the random part of the spectrum, which affects the finality of certain trajectories. As in our previous investigations on matrices, among all the IR properties, we focus on the vacuum expectation value.

Furthermore, based on a sixtic truncation using the parameterization (Equation 28), we show in Figure 5 an explicit realization of this change in asymptotic behavior. We observe that the RG trajectory ends in the symmetric phase in the case of pure noise (on the left) and stays in the non-symmetric phase when we add a signal (on the right). The symmetry restoration observed for the purely noisy signal is lost when a strong enough signal is added. The set of initial conditions in the vicinity of the Gaussian fixed point that ends in the symmetric phase form a compact region that we call $R_{0}$; as pictured in Figure 6. Moreover, adding a signal reduces the size of this region, and this behavior already observed for matrices has been observed in all the situations that we have been able to investigate numerically. Finally, the same conclusions hold in the non-symmetric phase, expanding the effective potential around a running vacuum κ as in (Equation 14) using LPA. Figure 6 provides an illustration of the corresponding region $R_{0}$. Note that, as explained in the previous section and as it is the case for matrices, the anomalous dimension does not play a relevant role, and no significant difference has been observed using LPA or LPA${}^{\prime }$. Finally, let us mention an important point, already discussed for matrices in [24]. All the trajectories into the region $R_{0}$ are not physically relevant. Indeed, from our interpretation, the effective mass in the deep IR must correspond to the inverse of the largest eigenvalue of the covariance matrix. Thus, only a subset of $R_{0}$ is physically relevant. This suggests again the existence of an intrinsic detection threshold. The deformation of the region $R_{0}$ due to the signal must be sufficient to affect physically relevant trajectories.

## 4. Conclusions and Open Issues

In this paper, we investigated some aspects of the intrinsic RG flow encoded from coarse-graining of the eigenvalue distribution for a suitable generalization of the covariance matrix based on the elementary melonic invariant. We have thus provided some strong evidence in favor of the universality of the behavior observed in nearly continuous spectra around Marchenko–Pastur law, that we recalled in the introduction. In particular:(i)We showed that the intrinsic RG flow associated with a single i.i.d random tensor has only a few numbers of relevant local interactions and that the dimension of the relevant sector is essentially the same (restricting ourselves to the local approximation) for all the realizations at large *N*. Thus, assuming that the properties of a purely noisy data set can be well materialized by such an i.i.d random tensor. We showed that the presence of a strong enough signal (suitably materialized by a sum of discrete spikes) reduces the dimension of the relevant sector, modifying accordingly the properties of the asymptotic distributions for the (effective) random field.(ii)For the purely random distributions defining the noise, we showed the existence of a simply connected compact region $R_{0}$ in the vicinity of the Gaussian fixed point, for which the $Z_{2}$-symmetry is always restored in the deep infrared; the RG trajectories end in the symmetric region with $\bar{\phi }=0$. Moreover, disturbing a spectrum with a strong enough signal systematically reduces the size of this compact region, stressing a link between signal detection and $Z_{2}$-symmetry breaking. In this picture, we expect that the strength of the signal plays an analogous role to the “temperature” [24] in the standard critical phenomena description. Furthermore, only a subregion of $R_{0}$ provides physically relevant states in the infrared regime. We thus conjecture the existence of an intrinsic detection threshold since it is expected that the signal can only be detected when the physical sub-region of $R_{0}$ is affected by the deformation.

These conclusions depend on the simplifications that we did in our analysis, and we plan to improve these challenging issues in our future work. This especially concerns two aspects. The first one is about our restriction to the elementary melon in order to define the 2-point correlations. As the authors in [37] showed, it may be relevant to consider other definitions, based on non-melonic tensorial invariants, as illustrated in Figure 1. The second source of approximation concerns the procedure used to solve the RG Equation (Equation 12). Indeed, for simplicity, we chose to work within the LPA approximation, and thus to focus on the first terms in the derivative expansion. We have reason to believe that such an approximation works well for investigations on the tail of the spectrum (in deep IR limit). These corrections could play a significant role in signing the presence of the signal—improving our understanding of the meaning of our effective objects like the field $\phi_{i}$—without modifying qualitatively our conclusions. However, more sophisticated methods could be required for investigations when one moves away from this zone, for eigenvalues of intermediate size [56,57]. The third source of approximation concerns the field theoretical embedding itself. As discussed in the introduction, in absence of a fundamental understanding, this embedding is only expected to provide an effective description of the correlations into data sets, able to capture relevant features coming from coarse-graining of the eigenvalue distribution and reproduce partially the most relevant statistical features of the data sets. However, we notice that this field theory approximation seems to be more solid for tensors than for matrices. Indeed, for matrices, we showed that the dimension of the relevant sector (spanned with local observables having positive canonical dimension) increases arbitrarily toward UV scales. In contrast, the relevant sector for tensors contains only a few numbers of local observables, for a large part of the spectrum. Moreover, the dimension of the relevant sector never becomes large; on the contrary, it tends to be reduced to the Gaussian term in the ultraviolet limit. Theoretical embedding, which is not field theory, has been suggested in [23] and expected to be another relevant direction of investigation for future works.

## Figures and Tables

**Figure 1 entropy-23-00795-f001:**
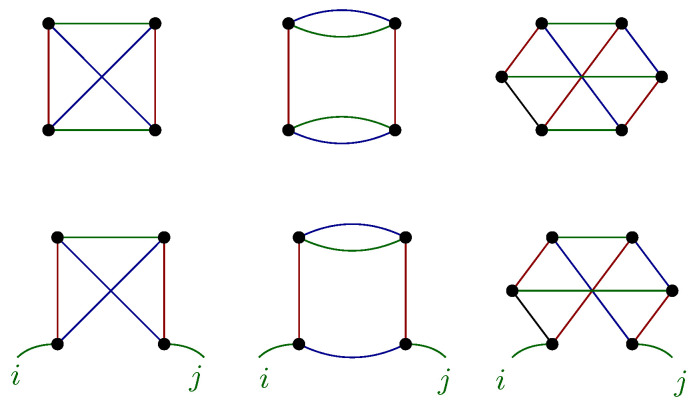
Illustration of some tensorial invariants for tensors of rank 3 with the graphical representation (on the **top**), and the corresponding generalizations of the covariance matrix (on the **bottom**).

**Figure 2 entropy-23-00795-f002:**
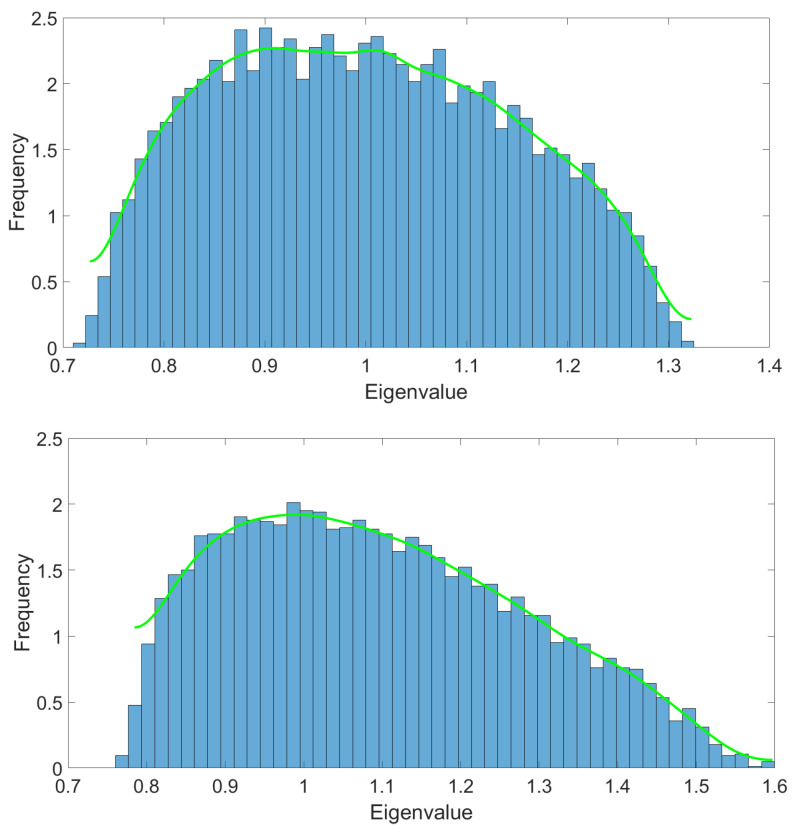
Typical eigenvalue spectra corresponding to a purely random tensor (on the **top**) and the same data with some spikes added to it. The first histogram is obtained from 100 realizations of the eigenvalues distribution associated to i.i.d random tensors of rank 3 and size *N*^3^ = 50^3^. The second picture (in the **bottom**) is constructed from the superposition of the random tensor with 50 (suitably normalized) spikes. Finally, the green curve materializing the numerical interpolation.

**Figure 3 entropy-23-00795-f003:**
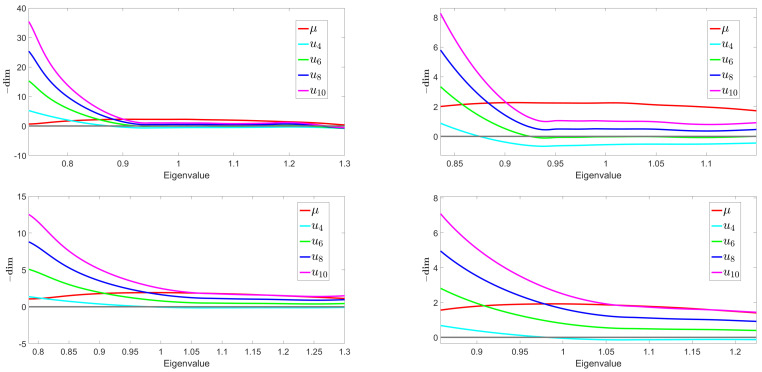
Opposite of the canonical dimensions (**top**) and the canonical dimensions associated to the eigenvalue distribution (**bottom**).

**Figure 4 entropy-23-00795-f004:**
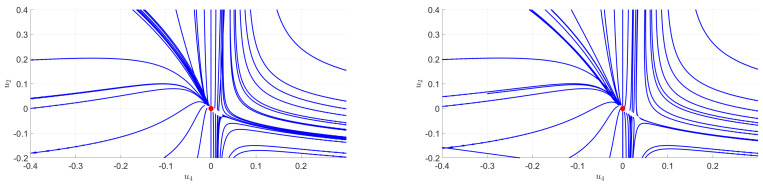
Numerical flow associated to the eigenvalue distribution of the generalized covariance matrix of a purely random tensor without signal (**left**) and with signal (**right**) for the quartic truncation (the arrows being oriented from UV to IR).

**Figure 5 entropy-23-00795-f005:**
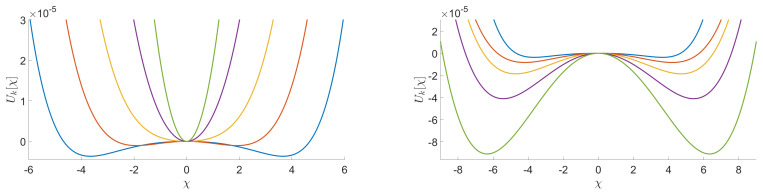
Illustration of the evolution of the potential associated to the couplings $u_{2}$, $u_{4}$, and $u_{6}$ for a truncation around χ=0. This example corresponds to specific initial conditions (in blue). We illustrate different points of the trajectory, from UV to IR respectively by the blue, red, yellow, purple, and green curves. We observe that this RG trajectory ends in the symmetric phase in the case of pure noise (on the **left**) and stays in the non-symmetric phase when we add a signal (on the **right**).

**Figure 6 entropy-23-00795-f006:**
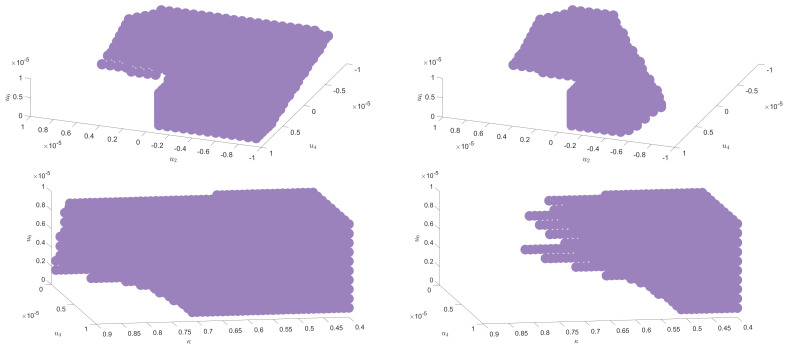
Illustration of the compact region $R_{0}$ (illustrated with purple dots) in the vicinity of the Gaussian fixed point providing initial conditions ending in the symmetric phase. On the **left**: the region for purely i.i.d random tensors in the expansion around χ=0 (on the **top**) and around a running vacuum χ=κ (on the **bottom**). On the **right**: the same regions when a signal build as a sum of discrete spikes is added.

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
