# Peer review of "Field Theoretical Approach for Signal Detection in Nearly Continuous Positive Spectra II: Tensorial Data"

_entropy, 2021, doi:10.3390/e23070795_

Round 1

Reviewer 1 Report

Summary of the paper: Tensorial principal component analysis is a generalization of principal component analysis that focuses on data that is better expressed by tensors rather than matrices. This paper investigates signal detection for the difficult problem of nearly continuous spectra using the nonperturbative renormalization group formalism based on a minor generalization of the covariance matrix. The renormalization group allows effective descriptions to be constructed while retaining only relevant features in the large eigenvalues, resulting in universal descriptions that enable the presence of the signal to be correlated with objectives and computable quantities. The vacuum expectation value is one of them, and we will concentrate on it in this article. The authors present experimental evidence in support of a connection between symmetry breaking and the existence of an intrinsic detection threshold.

Evaluation: I am convinced by the importance of the proposed methodology, and the quality of the paper is acceptable. I however formulate the following comments that need to be taken into account before a decision:   

o Define clearly all the introduced mathematical quantities (tilde C, etc.); all must be clear for the reader.

o The captions of the figures  are a too massive; I have rarely seen that. Normally, the details are described in the text, and the captions present the figures at the minimum of detail.

o Most sums have no upper bound. Please add the limits in the sum for the sake of clarity in the formulas.

o Most of the 2020 references are those of the authors. Can we have some recent references (2020 - 2021) on the subject with comments beyond the own authors work? I mean, it is to show the actual importance of the topic to the reader, beyond the own authors vision (this remark is not formulated negatively). 

o I am not sure that the concept of Entropy, at the center of the journal, is mentioned somewhere. 

o The English must be improved a bit before a possible publication.

Reviewer 2 Report

Signal detection in nearly continuous spectra is a challenging problem for which the renormalization group approach is an effective method. This paper, as a continuation of [22-24], proposes to use an effective field theoretical embedding able to partially reproduce the correlations to address some issues in TPCA through a suitable coarse-graining procedure to construct renormalization group flow. This paper has several solid technical contributions, and is well written and clearly presented.

Below are some minor comments to help the authors improve the readability of the paper.
Section I:
"to extract of" should be "to extract".
The symbol $^{\otimes k}$ is not defined.
"a N-components field" should be "an N-component field".

Section II.A:
"As explain" should be "As explained".
For self-containedness of the paper, it needs to define the symbol $O(\cdot)$ and briefly explain what is $O(P)$-invariant.
The formula of Eq. (2) is unclear to me. Especially, what does the symbol $\times$ mean?
In FIG. 2 caption, "materializing" should be "materializes".
"Figure 2 illustrate(s)".
"with a signal build (built) as a sum of spikes".
"the two matrices are certainly equals" should be "... are ... equal".
In the fourth line below Eq. (10), does "the experimental distribution" mean "the empirical distribution"?

Round 2

Reviewer 1 Report

The revised version takes into account my remarks. The present version is much better and at the level of a high quality journal. I recommend it for publication.